# Real-Time Coaching of Human Physical Skills with Large Language Models

## Abstract

Concurrent coaching of humans with language instruction has the potential to dramatically accelerate skill acquisition in high-stakes domains like driving and sports. However, effective concurrent coaching requires two key capabilities: determining *when* to intervene with fast, proactive timing decisions, and determining *what* to say through free-form instruction generation for diverse scenarios. Existing approaches struggle because they either sacrifice real-time responsiveness for content quality or sacrifice content flexibility for speed. Our key insight is to decompose concurrent coaching into two stages: deciding *when* to intervene and determining *what* to say, bridged by a shared representation. We introduce STREAMCOACH, a two-stage coaching framework that encodes learner state into lightweight language embeddings, enabling intervention decisions within 17 ms that trigger generation of contextually appropriate instructions. In the fast inference stage, STREAMCOACH compares current state embeddings against past coaching scenarios to trigger interventions. In the slow reasoning stage, the same embeddings retrieve relevant examples for Retrieval-Augmented Generation of adaptive instructions. By separating timing-critical decisions from content generation, STREAMCOACH achieves both key capabilities simultaneously. Evaluated in high-performance driving, STREAMCOACH significantly outperforms existing approaches in both intervention timing and instruction quality, demonstrating effective concurrent coaching of humans through language.

## 1 Introduction

Concurrent coaching with language instruction, where coaches provide real-time guidance to humans during task execution, is a powerful tool for accelerating human skill acquisition (Magill & Anderson, 2017). A human driving coach, for example, might say "brake earlier here" or "steer tighter around this corner" to help a human learner adjust their technique on the fly. Unlike *terminal coaching*, which provides feedback to humans only after task completion when intervention opportunities are lost (see Figure 1), concurrent coaching offers immediate, context-aware guidance that helps humans prevent errors as they unfold (Hattie & Timperley, 2007; Denys Brand & Tortolero, 2020; Hula et al., 2008; Hodges & Williams, 2012). These timely interventions are especially critical for human performance in high-speed, high-stakes domains where delayed feedback arrives too late to help humans adjust their actions (Gopinath et al., 2025). Automating such human coaching with AI could dramatically expand access to expert feedback and provide personalized support for human learners where human coaches are unavailable.

However, effective concurrent human coaching with AI systems requires two key capabilities (see Table 1): determining *when* to intervene with fast, proactive timing decisions, and determining *what* to say through free-form instruction generation for diverse scenarios and human learner behaviors. Terminal approaches like CORGI (Srivastava et al., 2023) generate quality instructions but operate post-task, causing learners to repeat errors without timely correction. Conversational systems like GPTCoach (Jörke et al., 2025) provide flexible dialogue but remain passive, missing critical intervention opportunities while waiting for user queries. Concurrent systems like Gopinath et al. (2025) achieve fast timing but use fixed rule sets, potentially providing inappropriate guidance in novel situations. End-to-end approaches like Panchal et al. (2024) attempt both capabilities in a single vision-language model but struggle with timing complexity, causing learners to miss intervention windows during slow joint optimization.

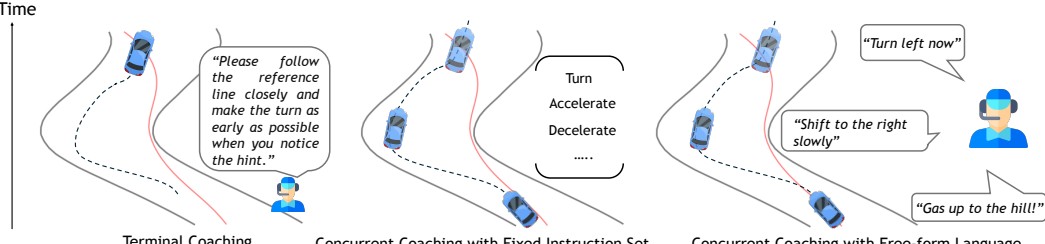

Figure 1: **Comparison of Coaching Methods**: Terminal coaching, where feedback is given after the session for overall performance improvement; Concurrent coaching with fixed instructions, which provides real-time guidance using structured commands; and Concurrent coaching with free-form language. The gray lines represent the track borders, while the red line illustrates the reference driving line for the optimal path. The dotted line indicates the actual driving trajectory of the driver.

Table 1: **Comparison of Coaching Systems:** Two key capabilities for effective concurrent coaching. **When to intervene**: *Concurrent* (provides guidance during task execution), *Fast Intervention* (makes timing decisions <100ms), *Proactive* (actively determines intervention moments vs. waiting for queries). **What to say**: *Free-form* (generates flexible instructions for diverse scenarios).

| Coaching System | Concurrent | Fast Intervention | Proactive | Free-form |
|---|:---:|:---:|:---:|:---:|
| CORGI (Srivastava et al., 2023) | — | — | — | ✓ |
| Gopinath et al. (2025) | ✓ | ✓ | — | — |
| GPTCoach (Jörke et al., 2025) | — | — | — | ✓ |
| Panchal et al. (2024) | ✓ | — | ✓ | ✓ |
| **STREAMCOACH (Ours)** | ✓ | ✓ | ✓ | ✓ |

To address these challenges, our main insight is to decompose concurrent coaching into two stages: deciding *when* to intervene and determining *what* to say, bridged by a shared representation. We introduce STREAMCOACH, a two-stage coaching framework inspired by (Sinha et al., 2024), which pairs fast inference for intervention timing with slow reasoning for instruction generation, as illustrated in Figure 2. In the fast inference stage, STREAMCOACH encodes the learner's real-time state, including actions, trajectories, and environmental cues, into lightweight language embeddings (Reimers & Gurevych, 2019), continuously comparing these against past coaching scenarios with expert feedback. Critically, STREAMCOACH can determine whether to intervene within 17ms, meeting fast intervention requirements.

Crucially, this same embedding space powers the content generation stage: once intervention is triggered, the similarity scores are used to retrieve relevant prior coaching episodes, which are then used in a Retrieval-Augmented Generation (RAG) pipeline. These examples ground a language model to compose tailored, domain-specific instructions, thus bypassing the need for densely labeled training data, unlike prior systems. The shared embedding acts as a bridge between timing and content, enabling efficient, consistent, and context-aware coaching.

In this work, we explore the application of STREAMCOACH for concurrent coaching in high-performance driving (Betz et al., 2022; Wurman et al., 2022; Werner et al., 2023; Chen et al., 2023; DeCastro et al., 2024; Gopinath et al., 2025), with a focus on evaluating the timing and quality of the generated instructions. Our results show that STREAMCOACH delivers accurate, contextually relevant guidance with fast intervention timing, outperforming baselines in both intervention timing and instruction quality. By unifying fast intervention detection and slow instruction generation through a shared embedding space, STREAMCOACH enables scalable, concurrent language-based coaching.

## 2 RELATED WORK

**LLMs for Education.** LLMs offer personalized and scalable learning experiences through natural language interaction (Xu et al., 2024; Wang et al., 2024a). They have been applied to problem-solving (Wu et al., 2023b; Bommarito II & Katz, 2022; Cui et al., 2023b; Liévin et al., 2023; Thirunavukarasu et al., 2023; Wu et al., 2023a; Yang et al., 2023; Kazemitabaar et al., 2023; Savelka et al., 2023; OpenAI, 2023; Zhang et al., 2024), error correction (Zhang et al., 2023; Zhao et al., 2023), question generation (Doughty et al., 2024; Lee et al., 2023; Xiao et al., 2023), etc. Fine-tuning on domain-specific data enhances their pedagogical alignment, yet most applications target conceptual

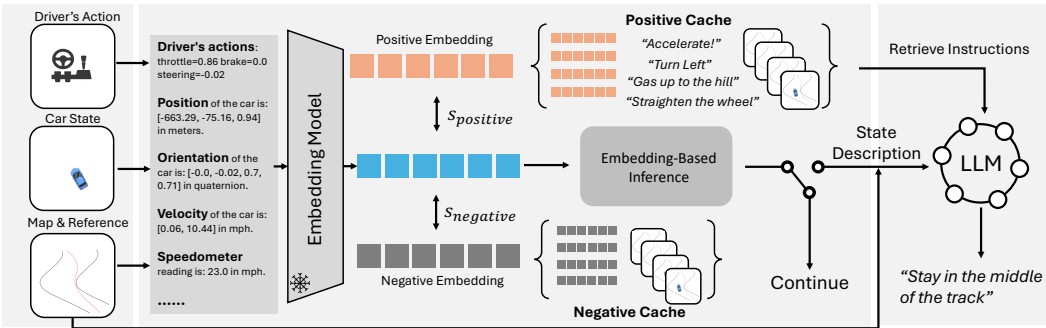

Figure 2: **Overview of STREAMCOACH during Inference Time. Left:** STREAMCOACH converts the current driver action, car state, and map data into a language description. **Middle:** The description is embedded using a language model. Cosine similarity with cached positive/negative embeddings and a trained classifier determine whether to trigger slow reasoning. **Right:** If triggered, relevant instructions are retrieved from the positive cache together with the state description for retrieval-augmented generation.

tasks rather than physical skills. Teaching physical skills requires LLMs to interpret multimodal inputs and actions. A pioneering work has used LLMs for terminal feedback (Srivastava et al., 2023), but this approach only offers post-task evaluation. In contrast, concurrent teaching requires real-time, context-sensitive guidance that allows learners to adjust their actions on the fly. In this work, we present STREAMCOACH, a model that generates immediate, precise instructions through a fast-slow inference framework for real-time skill learning.

**LLMs for Autonomous Driving.** LLMs are being explored in autonomous driving to enhance high-level reasoning—such as interpreting traffic laws, generating behavior strategies, and assisting with path planning (Shao et al., 2024; Wang et al., 2023; Mao et al., 2023b;a; Sima et al., 2024). They also improve human-vehicle interaction by enabling natural language commands and are used in retrieval-augmented systems to explain agents' behaviors (Yuan et al., 2024; Hussien et al., 2024; Cui et al., 2023a; 2024a; Ma et al., 2024; Cui et al., 2024b). Unlike these applications that generate vehicle behavior, our work focuses on producing timely instructional feedback for human learners. Rather than replicating expert driving behavior, STREAMCOACH observes and analyzes the learner's actions to provide corrective guidance that promotes proper technique and decision-making.

**Retrieval Augmented Generation.** Retrieval Augmented Generation integrates LLMs with external retrieval mechanisms to enrich generation with domain-specific knowledge (Gupta et al., 2024; Li et al., 2025; Rau et al., 2024; Wang et al., 2024b; Zhao et al., 2024; Shen et al., 2024; Han et al., 2025; Li et al., 2024; Gao et al., 2024; Lewis et al., 2020). By querying a curated repository during inference, RAG incorporates relevant examples or expert annotations, leading to more informed responses (Yuan et al., 2024; Hussien et al., 2024). In our framework, the slow reasoning stage employs RAG to retrieve relevant experiences and generate nuanced, context-aware instructions.

## 3   PROBLEM FORMULATION

We treat concurrent teaching as a sequential process in which the system operates at discrete time steps $t = 1, 2, \ldots, T$. At each time $t$, the system observes inputs $o_t = \{o_t^{\text{state}}, o_t^{\text{behavior}}, o_t^{\text{task}}\}$, where $o_t^{\text{state}}$ captures the current state of the environment (e.g. position of the car), $o_t^{\text{behavior}}$ represents the human's ongoing behavior (e.g., brake), and $o_t^{\text{task}}$ encodes task-specific information such as map information, and produces a free-form instruction $\mathcal{I}_t \in \mathcal{L} \cup \{\emptyset\}$, where $\emptyset$ indicates that no instruction is given and $\mathcal{L}$ is the language instructions can be generated. Our goal is to generate well-timed instructions with content closely aligned to expert instructions. To quantify this, we define a teaching score at each step, $R_t = r_t^{\text{timing}} * r_t^{\text{content}}$ and consider the total score over the whole task horizon as the overall measure of teaching quality. We assume access to a dataset $\mathcal{D} = \left\{ \left( o_{1:T}^{*(i)}, \mathcal{I}_{1:T}^{*(i)} \right) \right\}_{i=1}^{N}$, where $\mathcal{I}_t^* = \emptyset$ indicates that no expert instruction was given at time $t$. The timing score $r_t^{\text{timing}}$ evaluates whether the generated instruction $\mathcal{I}_t$ (when it is not $\emptyset$) is issued within a valid interval $[t_{\text{start}}, t_{\text{end}}]$ computed with respect to the timing of the ground truth instruction:

$$r_t^{\text{timing}} = \begin{cases} 1, & \text{if } t_{\text{generation}} \in [t_{\text{start}}, t_{\text{end}}], \\ 0, & \text{otherwise.} \end{cases} \quad (1)$$

Here, $t_{\text{generation}}$ denotes the time step at which the model issues the instruction $\mathcal{I}_t$. The content score $r_t^{\text{content}}$ measures the similarity between the generated instruction $\mathcal{I}_t$ and the expert instruction $\mathcal{I}_t^*$. Common metrics include cosine similarity or BLEU/ROUGE (Papineni et al., 2002; Lin, 2004):

$$r_t^{\text{content}} = \text{sim}(\mathcal{I}_t, \mathcal{I}_t^*).$$

The teaching strategy should yield a high average score across the entire duration of human training. Over the dataset $\mathcal{D}$, the system aims to learn a mapping from past observations $o_{1:t}$ to instructions $\mathcal{I}_t$ that maximizes this measure of both timely and relevant feedback.

### 3.1 Task Domain:
### Concurrent Coaching for High Performance Driving

High performance driving is a dynamic, high-stakes environment where split-second decisions and precise maneuvers are crucial. This work focuses on the domain of concurrent coaching for high performance driving, where the goal is to deliver real-time, actionable feedback that enables drivers to adjust their driving technique in real time. We use CARLA (Dosovitskiy et al., 2017) as the simulation platform and adopt the Thunderhill West track (Willows, CA) as the driving circuit; see Figure 3 for an illustration.

Figure 3: **Map of the Racing Track.** The green stars represent cones placed along the lap to mark key points, the grey lines indicate the track, and the red line illustrates the reference driving line for optimal driving path.

## 4 StreamCoach

To address these challenges, we propose a fast-slow inference framework. StreamCoach operates in two key stages, as illustrated in Figure 2 and algorithm 1.

### 4.1 Fast Inference

Fast inference serves as the first stage in our fast-slow framework, quickly assessing the need for intervention. It combines precomputed language embeddings with a task-specific classifier, enabling efficient, context-aware decisions. The hybrid approach balances speed and adaptability: embeddings support rapid semantic matching, while the classifier handles subtle variations for robust performance.

To enable embedding-based reasoning, each observation $o_t \in o_{1:T}$ is first converted into a natural language description using templates (Appendix H), similar to Hwang et al. (2024) and Sinha et al. (2024). These templates extract key features from the system state, task objectives, and learner behavior, and transform them

---

**Algorithm 1** StreamCoach

1: **Input:** Observation $o_t$, embedding function $\phi$, positive cache $D_{\text{positive}}$, negative cache $D_{\text{negative}}$, classifier $f$, threshold $\tau$, retrieval parameter $k$, RAG model.
2: **Output:** Instruction $\mathcal{I}_t$ or no instruction.
3: $\triangledown$ *Fast Inference Stage*
4: Compute embedding: $e_t \leftarrow \phi(o_t)$
5: Compute similarity scores:
$$s_{\text{pos}} \leftarrow \max_{e^* \in D_{\text{positive}}} \frac{e^{*\top} e_t}{\|e^*\|\|e_t\|},$$
$$s_{\text{neg}} \leftarrow \max_{e^* \in D_{\text{negative}}} \frac{e^{*\top} e_t}{\|e^*\|\|e_t\|}$$
6: Compute decision score: $\Delta s \leftarrow (s_{\text{pos}} - s_{\text{neg}})$
7: **if** $\Delta s < 0$ **and** $f(e_t) = 0$ **then**
8:    Return $\mathcal{I}_t = \emptyset$
9: **end if**
10: $\triangledown$ *Slow Reasoning Stage*
11: Retrieve top-$k$ similar experiences: $\mathcal{E}_t \leftarrow$ Top-$k \left\{ e^* \in D_{\text{pos}} : \frac{e^{*\top} e_t}{\|e^*\|\|e_t\|} \right\}$
12: Retrieve corresponding instructions for each $e^* \in \mathcal{E}_t$
13: Generate instruction: $\mathcal{I}_t \leftarrow \text{RAG}(\mathcal{E}_t, o_t)$
14: Return $\mathcal{I}_t$

---

into structured, natural language statements that preserve essential contextual information. For notational simplicity, we continue to denote these text-based representations as $o_t$, with the understanding that they refer to the language descriptions derived from raw observations.

Given a training dataset $\mathcal{D} = \left\{ \left( o_{1:T}^{*(i)}, \mathcal{I}_{1:T}^{*(i)} \right) \right\}_{i=1}^{N}$, each observation $o_t^* \in o_{1:T}^{*(i)}$ is mapped to an embedding $e_t^* \in \mathbb{R}^d$ using an off-the-shelf pretrained language embedding model $\phi(\cdot)$ (Song et al., 2020). These embeddings are then partitioned into:

$$D_{\text{positive}} = \{e_t^* \mid \mathcal{I}_t^* \neq \emptyset\}, \quad D_{\text{negative}} = \{e_t^* \mid \mathcal{I}_t^* = \emptyset\}, \tag{2}$$

which enables the system to differentiate between scenarios that require intervention ($D_{\text{positive}}$) and those that do not ($D_{\text{negative}}$). Together, $D_{\text{positive}}$ and $D_{\text{negative}}$ constitute the *embedding retrieval* cache.

Fast inference determines whether to generate an instruction by leveraging prior experiences stored in the *embedding retrieval* cache. Conversely, all other scenarios are categorized as negative scenarios.

While embeddings effectively capture general semantic meanings, they may not adequately represent task-specific patterns. For example, two observations involving a sharp left turn and a gentle left curve may yield similar embeddings due to shared lexical cues, despite requiring different instructions and intervention strategies. This semantic overlap can lead to ambiguous or suboptimal guidance if the retrieval mechanism lacks sensitivity to task-relevant nuances such as motion dynamics.

To enhance embedding-based reasoning, we train a binary classifier that maps language embeddings to an instruction occurrence indicator: $f : \mathbb{R}^d \to \{0, 1\}$, where the output denotes whether an instruction was issued (1) or not (0) for a given state embedding. Although the classifier improves task-specific adaptability, it may compromise some of the broader semantic information inherent in the embeddings due to in-domain fine-tuning (Kotha et al., 2024; Luo et al., 2023). To address this, we implement a hybrid decision strategy that combines the classifier's output with embedding-based similarity comparisons. At runtime, the embedding $e_t = \phi(o_t)$ for a new observation is computed. Its similarity to both $D_{\text{positive}}$ and $D_{\text{negative}}$ is measured using cosine similarity:

$$s_{\text{positive}}(e_t) = \max_{e^* \in D_{\text{positive}}} \frac{e^{*\top} e_t}{\|e^*\| \|e_t\|}, \quad s_{\text{negative}}(e_t) = \max_{e^* \in D_{\text{negative}}} \frac{e^{*\top} e_t}{\|e^*\| \|e_t\|}. \tag{3}$$

The final decision combines this score with the classifier's prediction:

$$\mathcal{I}_t = \begin{cases} \text{generate instruction}, & \text{if} \quad s_{\text{positive}}(e_t) < s_{\text{negative}}(e_t) \quad \textbf{or} \quad f(e_t) = 1, \\ \emptyset, & \text{otherwise.} \end{cases} \tag{4}$$

For all states within the time window $[t_{\text{start}}, t_{\text{end}}]$, we classify them as a positive scenario, assuming the entire window shares the same instruction. This hybrid approach ensures that instructions are generated based on either a stronger similarity to the most positive experience compared to the most negative experience or the classifier's positive prediction. In practice, we use two frames of observation, the current frame and the previous frame, to extract embeddings (see Appendix H for details). For clarity of notation, this detail is omitted in the equations.

## 4.2 Slow Reasoning

Slow reasoning refines the decision-making process by leveraging the embedding generated during fast inference to retrieve relevant past experiences and generate a contextually appropriate free-form instruction $\mathcal{I}_t$.

We denote the reasoning model as $\mathcal{R}$, which maps a composite prompt $P_t$ to a free-form instruction $\mathcal{I}_t$. Given the current observation $o_t$, its embedding $e_t = \phi(o_t)$ is computed during fast inference. This embedding is used to retrieve a set of relevant past experiences from the positive cache $D_{\text{positive}}$. Specifically, the retrieval set

$$\mathcal{E}_t = \text{Top-}k\left( \frac{e_t^{*\top} e_t}{\|e_t^*\| \|e_t\|} \mid e_t^* \in D_{\text{positive}} \right) \tag{5}$$

is constructed by selecting the top $k$ embeddings $e_t^*$ with the highest cosine similarity to $e_t$. Each retrieved embedding $e_t^* \in \mathcal{E}_t$ is linked to its historical instruction $\mathcal{I}_t^*$ from the dataset $\mathcal{D}$, providing contextually relevant instruction examples to inform the generation of $\mathcal{I}_t$.

The retrieved instruction-embedding pairs $\{(e_t^*, \mathcal{I}_t^*)\}_{e_t^* \in \mathcal{E}_t}$ serve as the basis for constructing the composite prompt $P_t$ (Appendix I and F.2). The prompt $P_t$ integrates these retrieved examples with additional contextual details from the current observation $o_t$. The reasoning model $\mathcal{R}$ then processes the prompt $P_t$ to generate a new instruction: $\mathcal{I}_t = \mathcal{R}(P_t)$, ensuring that the generated instruction is both semantically aligned with historical examples and adapted to the current context.

We present two ways to implement the RAG model (i.e., the reasoning model $\mathcal{R}$) for slow reasoning:
**Prompting-Based Approach.** A large pretrained LLM is used as is. This approach is straightforward to deploy and requires no additional training, making it flexible and easily adapted to new tasks.
**Fine-Tuned Approach.** In this variant, the LLM is further trained on the positive cache $D_{\text{positive}}$. Each training example is augmented with the top-$k$ retrieved instructions, incorporating them into the prompt $P_t$ during fine-tuning. Note in this variant, we encode the state $o_t$ directly into embeddings, rather than converting it into textual form (Appendix F.2 for details). This process teaches the model to leverage past examples and domain-specific context when generating new instructions.

## 5 EXPERIMENT

We aim to investigate the following questions in the experiment section: *RQ1*: Can our proposed framework accurately determine *when* to provide instructions in real time (i.e., timing) while learners perform dynamic tasks? *RQ2*: Does our approach generate instructions whose *content* aligns well with expert guidance across diverse scenarios? *RQ3*: How does our fast-slow inference framework compare to existing baselines in terms of both timing and content quality?

### 5.1 DATA CURATION

We collected the dataset from a study involving 15 participants who were instructed by an expert coach during a simulated high-performance driving task in CARLA (Dosovitskiy et al., 2017) on a single race track. This study was reviewed and approved by an IRB (name and details upon publication). Participants were given \$150 for their participation. Prior to participation, participants were given a consent form that outlined the risks of the study (potential motion sickness and eyestrain). After completing the consent form, the study began. Each study session lasted 2 hours. Participants drove with instruction from a professional driving coach. Every 15 minutes, participants took a short break. Subjects were instructed to listen to the coach and try their best to improve their lap time and racing-line adherence. The dataset includes 339 coaching trials sampled at 10Hz, resulting in 383,303 frames, covering 13,576 expert instructions after preprocessing. The model input $o_t$ consists of the following components: $o_t^{\text{state}}$, including *Position* ($\langle x, y, z \rangle$), *Velocity* ($\langle v_x, v_y \rangle$), and *Orientation* ($\langle o_x, o_y, o_z, o_w \rangle$) as quaternions; $o_t^{\text{behavior}}$, capturing the *Driver's Actions* ($\langle$Steering, Speedometer, Throttle, Brake$\rangle$); and $o_t^{\text{task}}$, which includes *Racing line and map information*, such as the nearest 20 coordinates on the reference optimal path and track borders relative to the current position. The dataset is divided into a training set (67%) and the remaining 33% for evaluation. The train-test split is based on different participants, ensuring no overlap and enabling robust testing on unseen individuals. More details can be found in Appendix G.

The data is collected from one expert coach to ensure consistency of the instructions for different students. By training on a single, highly-calibrated expert, we could isolate the core challenge of learning a coherent coaching policy before introducing the additional complexity of multi-expert disagreement. On average, the coach issues instruction every three seconds. We provide more qualitative demonstration of coach instructions in Appendix A.

### 5.2 BASELINES

**Baselines for Timing: 1. Classifier Only**: A neural network predicts binary outputs based on the embedding (details in Appendix F.3). **2. Embedding Only**: Instruction timing is determined by comparing the current state embedding to positive and negative embeddings in the retrieval pool: if the closest match is positive, an instruction is triggered; otherwise, it is not. **3. Rule-Based**: Manual rules trigger instructions when deviations from the optimal trajectory in position or velocity exceed predefined thresholds. We select the threshold that yields the best timing performance. **4. VideoLLM-Online** (Chen et al., 2024): processes streaming input by continuously outputting a special token to indicate no intervention and generates instructions only when necessary as in existing coaching system (Panchal et al., 2024). We adapt this model to take $o_t$ as input instead of images.
**Baselines for Content Evaluation:**
**Prompting-Based: 1. Zero-shot LLM**: Generates instructions directly from state descriptions without domain-specific examples. **2. Few-shot LLM**: Generates instructions using 30 in-domain examples (randomly selected from $D_{\text{positive}}$) for grounding as existing AI coaching system (Jörke et al., 2025). **3. Retrieval Top 1**: Retrieves the closest instruction from the training set via cosine similarity of observation embeddings.
**Fine-tuned Models: 1. Latest Observation LLM**: Generates instructions using only the latest 3 observations. **2. Full History LLM**: Similar to VideoLLM-Online, but generates instructions based on the full observation history without managing timing as in existing coaching system (Srivastava et al., 2023). **3. VideoLLM-Online**: Same as above.

### 5.3 IMPLEMENTATION DETAILS

STREAMCOACH can be implemented using two approaches, both leveraging MPNet (Song et al., 2020) for fast instruction retrieval via Sentence-Transformer (Reimers & Gurevych, 2019).

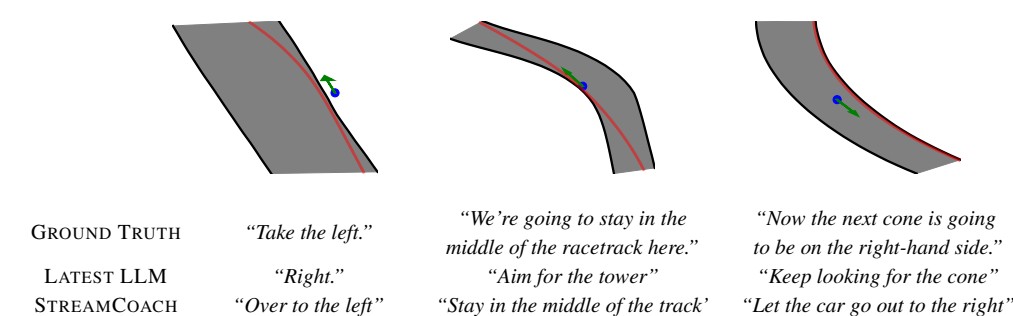

|  |  |  |  |
|---|---|---|---|
| GROUND TRUTH | *"Take the left."* | *"We're going to stay in the middle of the racetrack here."* | *"Now the next cone is going to be on the right-hand side."* |
| LATEST LLM | *"Right."* | *"Aim for the tower"* | *"Keep looking for the cone"* |
| STREAMCOACH | *"Over to the left"* | *"Stay in the middle of the track'* | *"Let the car go out to the right"* |

Figure 4: **Qualitative Results:** The blue dot represents the car's position, the green arrow shows its direction of movement, the red line marks the reference driving line, and the gray line outlines the track border. More qualitative results can be found in Appendix A.

The **prompting-based method** uses GPT-4o-Mini (OpenAI, 2023) without additional training. Current observations $o_t$ are converted into language descriptions along with context (past observations, task states, retrieved instructions) and input to the LLM for instruction generation using pre-trained reasoning abilities. The **fine-tuned method** trains LLaMa-3.1-8B-Instruct (Meta, 2024) on dataset $D_{\text{positive}}$ using LoRA (Hu et al., 2022), with three two-layer MLP encoders handling different $o_t$ input types, following LLaVA (Liu et al., 2023). We set $k = 30$ retrieved samples for prompting (Jin et al., 2024) and $k = 10$ for fine-tuned approaches. All results are averaged over three runs for stability. Our approach achieves real-time performance: 0.017s for embedding extraction and 0.54s/0.35s for instruction generation (prompting/fine-tuned) on A6000 GPU. These latencies are comparable to human instructor response times

While our state-based retrieval approach naturally adapts to changing driving scenarios, potential repetition within short time windows could occur in static situations. If needed, a simple temporal suppression mechanism comparing recent instruction embeddings could address this.

## 5.4 EVALUATION METRICS

The evaluation of STREAMCOACH focuses on two key aspects: *content similarity* and *timing*.

**Content Evaluation:** We measure content similarity $r_t^{\text{content}}$ using Cosine Similarity (Manning et al., 2008), BLEU-4 (Papineni et al., 2002), ROUGE (Lin, 2004), BERTScore (Zhang et al., 2020), and METEOR (Banerjee & Lavie, 2005) with embeddings from a paraphrase model (Wang et al., 2020). Additionally, GPT-4o (OpenAI, 2023) performs pairwise comparisons between generated instructions and ground truth $\mathcal{I}_t^*$, with randomized ordering to prevent bias.

**Timing Evaluation:** Timing accuracy $r_t^{\text{timing}}$ measures whether instructions occur within a 1.5-second window centered on expert timestamps. This stricter window (vs. 3-second in prior work (Panchal et al., 2024)) reflects real-time coaching demands. We report True Positive Rate (TPR), Balanced Accuracy, and $F_{\beta=2}$ Score.

**Overall Performance:** We define overall performance $R_t$ as the product of timing prediction accuracy (binary) and content similarity (cosine), capturing both decision-making and instruction quality in a unified metric.

## 5.5 EXPERIMENT RESULTS

The main content evaluation results are presented in Table 2 and Figure 5, with qualitative results in Figure 4. Zero-shot LLMs perform poorly due to their lack of task-specific knowledge, while few-shot LLMs, using limited in-domain examples, show improved performance by incorporating domain grounding. Methods like VideoLLM-Online, which handle both timing and content generation simultaneously, struggle to achieve both accuracy and contextual relevance. Embedding-based retrieval approaches perform well, as observation embeddings effectively capture task-relevant information. Even retrieving the top-1 instruction based on embeddings yields reasonable results, demonstrating their robustness for retrieval-augmented generation and domain-specific reasoning. Among all methods, STREAMCOACH achieves the best overall performance, with the fine-tuned version further improving results across all metrics by leveraging domain-specific training to achieve the highest scores and win rates.

Table 2: **Content Evaluation:** Generated Instruction Semantic Similarity Comparison. The table compares both prompting-based and finetuned-based approaches. CS stands for Cosine Similarity.

| | METHOD | CS | BLEU | BERTScore | METEOR | ROUGE |
|---|---|---|---|---|---|---|
| **PROMPTING** | ZERO-SHOT LLM | $0.2572_{\pm0.0062}$ | $0.0000_{\pm0.0000}$ | $0.8294_{\pm0.0007}$ | $0.0183_{\pm0.0029}$ | $0.0328_{\pm0.0040}$ |
| | FEW-SHOT LLM (JÖRKE ET AL., 2025) | $0.3204_{\pm0.0034}$ | $0.0206_{\pm0.0018}$ | $0.8627_{\pm0.0002}$ | $0.1620_{\pm0.0018}$ | $0.2209_{\pm0.0025}$ |
| | RETRIEVAL TOP 1 | $0.4168_{\pm0.0000}$ | $0.0545_{\pm0.0000}$ | $0.8721_{\pm0.0000}$ | $0.2186_{\pm0.0000}$ | $0.2747_{\pm0.0000}$ |
| | STREAMCOACH | $0.4512_{\pm0.0028}$ | $0.0927_{\pm0.0019}$ | $0.8766_{\pm0.0004}$ | $0.2769_{\pm0.0016}$ | $0.3352_{\pm0.0028}$ |
| **FINETUNED** | LATEST OBSERVATION LLM | $0.3116_{\pm0.0040}$ | $0.0395_{\pm0.0058}$ | $0.8680_{\pm0.0034}$ | $0.1696_{\pm0.0153}$ | $0.2333_{\pm0.0211}$ |
| | FULL HISTORY LLM (SRIVASTAVA ET AL., 2023) | $0.3277_{\pm0.0160}$ | $0.0431_{\pm0.0046}$ | $0.8671_{\pm0.0034}$ | $0.1557_{\pm0.0125}$ | $0.2132_{\pm0.0168}$ |
| | VIDEOLLM-ONLINE (CHEN ET AL., 2024) | $0.2280_{\pm0.0001}$ | $0.0056_{\pm0.0004}$ | $0.8368_{\pm0.0004}$ | $0.1240_{\pm0.0003}$ | $0.1343_{\pm0.0025}$ |
| | STREAMCOACH | $0.4966_{\pm0.0061}$ | $0.1017_{\pm0.0050}$ | $0.8879_{\pm0.0008}$ | $0.2908_{\pm0.0063}$ | $0.3746_{\pm0.0066}$ |

Table 3: **Timing Evaluation:** Timing performance of various models is evaluated using a 1.5-second time window.

| METHOD | TPR | ACCURACY | $F_{\beta=2}$ |
|---|---|---|---|
| CLASSIFIER ONLY | 0.5592 | 0.6213 | 0.5676 |
| EMBEDDING ONLY | 0.5513 | 0.5631 | 0.5465 |
| RULE-BASED | 0.3186 | 0.4353 | 0.3293 |
| VIDEOLLM-ONLINE (CHEN ET AL., 2024) | 0.0110 | 0.5029 | 0.0136 |
| STREAMCOACH | 0.7017 | 0.6133 | 0.6677 |

Table 4: **Overall Evaluation:** All models, except VideoLLM-Online, utilize the fine-tuned reasoning model from STREAMCOACH.

| TIMING MODEL | REASONING MODEL | TEACHING SCORE $R_t$ |
|---|---|---|
| CLASSIFIER ONLY | OURS | $0.2979_{\pm0.0033}$ |
| EMBEDDING ONLY | OURS | $0.2813_{\pm0.0038}$ |
| RULE-BASED | OURS | $0.2060_{\pm0.0032}$ |
| VIDEOLLM-ONLINE (CHEN ET AL., 2024) | | $0.0025_{\pm0.0000}$ |
| STREAMCOACH | | $0.3865_{\pm0.0049}$ |

Figure 5 presents the results of a head-to-head comparison using GPT-4o as a judge to evaluate the quality of generated instructions. In this evaluation, the fine-tuned version of StreamCoach is compared against three other models: the Latest Observation LLM, the Full History LLM, and the prompting-based version of StreamCoach. For each test case, the judge is presented with the ground-truth instruction and the instructions generated by both models in a randomized order.

Table 3 summarizes the timing performance for each method using a 1.5-second window. The Classifier Only method relies on task-specific features for binary predictions, while Embedding Only uses embeddings to compare the current state with positive/negative examples. Both achieve moderate performance but lack deeper task awareness. Our hybrid approach, combining these methods, achieves the best results overall by leveraging pretrained embeddings' semantic understanding and task-specific knowledge from the classifier. The Rule-Based method performs poorly, as expert instructions depend not only on deviations from a reference trajectory or velocity but also on the driver's performance.

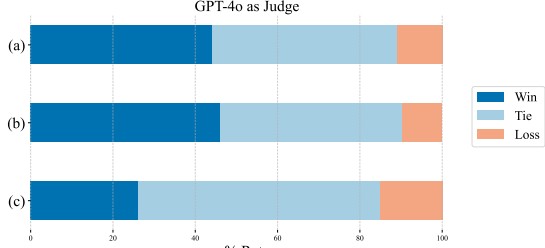

Figure 5: **LLM as Judge Results:** (a) STREAM-COACH (FT) vs. Latest State LLM, (b) STREAM-COACH (FT) vs. Full History State LLM, (c) STREAMCOACH (FT) vs. STREAMCOACH (Prompting). FT refers to the fine-tuned version of the reasoning model.

Table 4 presents the overall evaluation results, aligning with the standalone evaluations of timing and content. Leveraging the fast-slow framework, STREAMCOACH significantly outperforms methods that attempt to jointly learn timing and content, while maintaining real-time responsiveness.

### 5.5.1 ABLATION STUDY AND FURTHER ANALYSIS

We conduct ablation studies on the prompting-based variant for greater experimental flexibility.

**Retrieved Samples ($k$):** Performance stabilizes beyond $k = 30$ retrieved examples, indicating diminishing returns from additional context. While more examples provide diversity, excessive retrieval introduces noise in real-time systems. Figure 6 (a) shows this trade-off between contextual richness and accuracy.

**Time Window Size:** Larger windows improve performance by relaxing timing constraints but risk delayed feedback and overlapping instructions (expert intervals: 3 seconds). Small windows miss valid interventions due to human variability. We adopt 1.5 seconds as optimal: accommodating variability while maintaining responsiveness and tolerating generation latency ( 0.35s). Figure 6 (c)

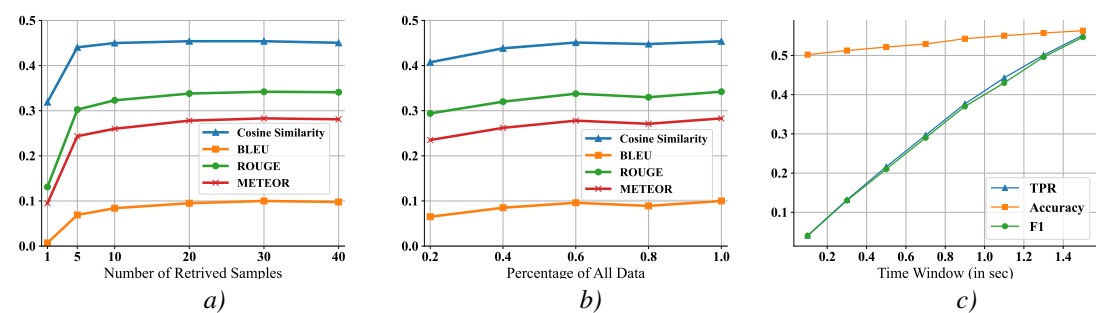

Figure 6: **Ablation Analysis:** *a*) Effect of the number of retrieved samples; *b*) Effect of retrieval cache size; *c*) Effect of time window size.

Table 5: **Ablation Study on Reasoning Models for Instruction Generation (Prompting Based):** All models use MPNet as the embedding model.

| METHOD | COSINE SIMILARITY | BLEU | BERTSCORE | METEOR | ROUGE |
|---|---|---|---|---|---|
| GPT-4O-MINI | $0.4512_{\pm 0.0028}$ | $0.0927_{\pm 0.0019}$ | $0.8766_{\pm 0.0004}$ | $0.2769_{\pm 0.0016}$ | $0.3352_{\pm 0.0028}$ |
| GEMINI 2.5 FLASH LITE | $0.4009_{\pm 0.0021}$ | $0.0663_{\pm 0.0019}$ | $0.8697_{\pm 0.0013}$ | $0.2176_{\pm 0.0063}$ | $0.2700_{\pm 0.0065}$ |
| CLAUDE 3.5 HAIKU | $0.4302_{\pm 0.0077}$ | $0.0758_{\pm 0.0068}$ | $0.8688_{\pm 0.0028}$ | $0.2493_{\pm 0.0151}$ | $0.2997_{\pm 0.0179}$ |

Table 6: **Ablation on Inference Speed vs. Retrieved Samples for Fine-tuned Models.**

| #$k$ | COSINE SIMILARITY | BLEU | BERTSCORE | METEOR | ROUGE | INFERENCE TIME (SECOND) |
|---|---|---|---|---|---|---|
| 1 | 0.4397 | 0.0662 | 0.8785 | 0.2259 | 0.3031 | 0.3168 |
| 5 | 0.4811 | 0.0937 | 0.8849 | 0.2702 | 0.3549 | 0.3343 |
| 10 | 0.4966 | 0.1017 | 0.8879 | 0.2908 | 0.3746 | 0.3503 |

demonstrates this trade-off.

**Retrieval Cache Size**: Performance stabilizes at 60% of training data, indicating that scenario diversity matters more than quantity. While larger caches provide broader scenario coverage, they increase retrieval complexity without proportional gains. Quality-diverse examples outweigh raw dataset size for effective retrieval.

**Inference Speed**: Inference time remains stable across different $k$ values due to short instruction length, while retrieval count significantly affects output quality. Table 6 shows this decoupling of speed and accuracy, enabling real-time performance without sacrificing instruction quality.

Table 7: **Slow Reasoning Time of Different Models.**

| METHOD | INFERENCE TIME (SECOND) |
|---|---|
| LATEST OBSERVATION LLM | 0.3728 |
| FULL HISTORY LLM | 1.6030 |
| VIDEOLLM-ONLINE | 0.8610 |
| STREAMCOACH | 0.3503 |

Our current implementation achieves 0.35s for content generation (reasoning time) on fine-tuned models with A6000 GPU, plus 0.017s for embedding extraction used in timing determination. This can be further accelerated using optimized inference frameworks like vLLM (Kwon et al., 2023). We also present the inference time of each fine-tuned model evaluated in the main paper in Table 7. Notably, for VideoLLM-Online, the reported time represents the total time required to determine when to intervene *and* generate instructions, as these processes are coupled rather than decoupled. Using full history as input increases inference time without significant performance gains, highlighting the effectiveness of our decoupled timing-content approach.

## 6 CONCLUSION

In this paper, we tackled the challenge of concurrent coaching for high performance driving using a fast-slow inference framework. Our approach combines quick decision-making with detailed, context-aware reasoning to generate clear and actionable free-form instructions. By using language embeddings and retrieval-augmented generation, the system integrates historical expert knowledge with the current context, ensuring timely and relevant feedback. We showed that our framework effectively balances the trade-offs between timing precision and content accuracy in demanding environments, even with limited annotated data. While our approach shows promise for real-time coaching, it has limitations. Although STREAMCOACH is a multimodal model that processes both tokenized state and language inputs, it lacks task-specific visual inputs such as driving scenes. Training vision-language models to handle such inputs would require large and diverse datasets, which are currently unavailable.

## REPRODUCIBILITY STATEMENT

To ensure reproducibility of our results, we provide comprehensive implementation details and experimental specifications throughout the paper and supplementary materials. The main paper includes detailed implementation information in Section 5.3, data collection procedures in Section 5.1, experimental setup in Section 5.4, and ablation studies in Section 5.5.1. Complete training and inference details, including hyperparameters, model architectures, and optimization settings for both prompting-based and fine-tuned approaches, are provided in Appendix F.1. The simulation environment setup and data collection protocol are described in Appendix G, while the complete prompt templates used for instruction generation are included in Appendix H.Appendix I contains the LLM-as-judge evaluation prompts and rubrics used for content quality assessment. Additional experimental details, including embedding model selections (Appendix C), inference time analysis (Appendix E), and extended qualitative results (Appendix A) are provided for comprehensive evaluation. All code, trained models, and datasets will be made available upon publication to facilitate reproduction and extension of this work.

## ETHICS STATEMENT

The STREAMCOACH is developed and evaluated exclusively within a simulated driving environment, with no direct or indirect control over real-world vehicles or physical systems. This strict simulation-only setup is central to our ethical positioning and ensures that the current work poses no physical, psychological, or safety risks to users.

**Controlled Setting and No Physical Actuation**   All instructions used to train STREAMCOACH are presented to human participants operating in a virtual car racing simulator. The system provides verbal feedback in natural language, but does not issue control commands or perform autonomous driving. This distinction is critical: the model operates purely as an assistive agent, with no actuation authority or embedded control loop with the environment. As such, there is no path from model output to real-world action that could lead to harm.

**Conservative Design for Instruction Timing and Content**   STREAMCOACH is designed with multiple safeguards that limit spurious or inappropriate interventions. The system only generates instructions when the current learner state closely matches expert-annotated examples from past data, based on embedding similarity and a trained classifier. In all other cases, it remains silent. Moreover, the use of retrieval-augmented generation ensures that the guidance provided is grounded in domain-relevant, expert-derived prior experience, rather than open-ended generation. This mitigates the risk of hallucinated or misleading instructions.

**No Ethical Risk from Data Use or Model Deployment**   The data used in this study was collected under Institutional Review Board (IRB) approval, with informed consent from all participants. The dataset contains no personally identifiable information, and is used solely for model training and evaluation in the simulator setting. The system is not deployed publicly, nor is it integrated into any real-world driving system or product. The entire pipeline—from data to evaluation—remains within a research sandbox, further limiting any potential downstream risks.

**Supportive, Not Prescriptive, Human-AI Interaction**   STREAMCOACH is fundamentally designed to support human learning, not to direct or override it. All generated instructions are suggestions, presented in natural language, and interpreted at the learner's discretion. There is no closed-loop automation or enforcement. This ensures that user agency is preserved and that learners remain in full control of the decision-making process throughout the task.

In conclusion, STREAMCOACH is ethically scoped by design. It operates in a simulated domain, with non-binding outputs, conservative intervention policies, and human-in-the-loop control. As such, the system introduces no new ethical risks in its current form, and we believe it provides a safe and responsible platform for exploring the potential of LLM-based real-time instruction.

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

## CONTENTS

## A  ADDITIONAL QUALITATIVE RESULTS

Here, we showcase more qualitative results in Figure 7.

## B  EXPERIMENTS WITH LLMS USING LONG-COT

Given the strong performance of reasoning-focused models on complex tasks, we evaluate one of the most capable publicly available models, *o4-mini* from OpenAI. Due to its relatively slow inference speed, we limit our evaluation to 10% of the total test set.

Our results in Table 8 indicate that incorporating Long Chain-of-Thought (Long-CoT) reasoning does not lead to performance gains. On the contrary, it introduces substantial computational overhead. Consequently, we exclude further experiments with Long-CoT in this work.

## C  ABLATION ON EMBEDDING MODEL SELECTIONS

Table 9: **Ablation on Embedding Models for Timing:** Performance of different embedding models is evaluated using the embedding-only method.

| METHOD | TPR | ACCURACY | $F_{\beta=2}$ |
|---|---|---|---|
| MPNET | 0.5513 | 0.5631 | 0.5465 |
| TEXT-EMBEDDING-3-SMALL | 0.5547 | 0.5603 | 0.5484 |
| TEXT-EMBEDDING-3-LARGE | 0.5473 | 0.5631 | 0.5435 |

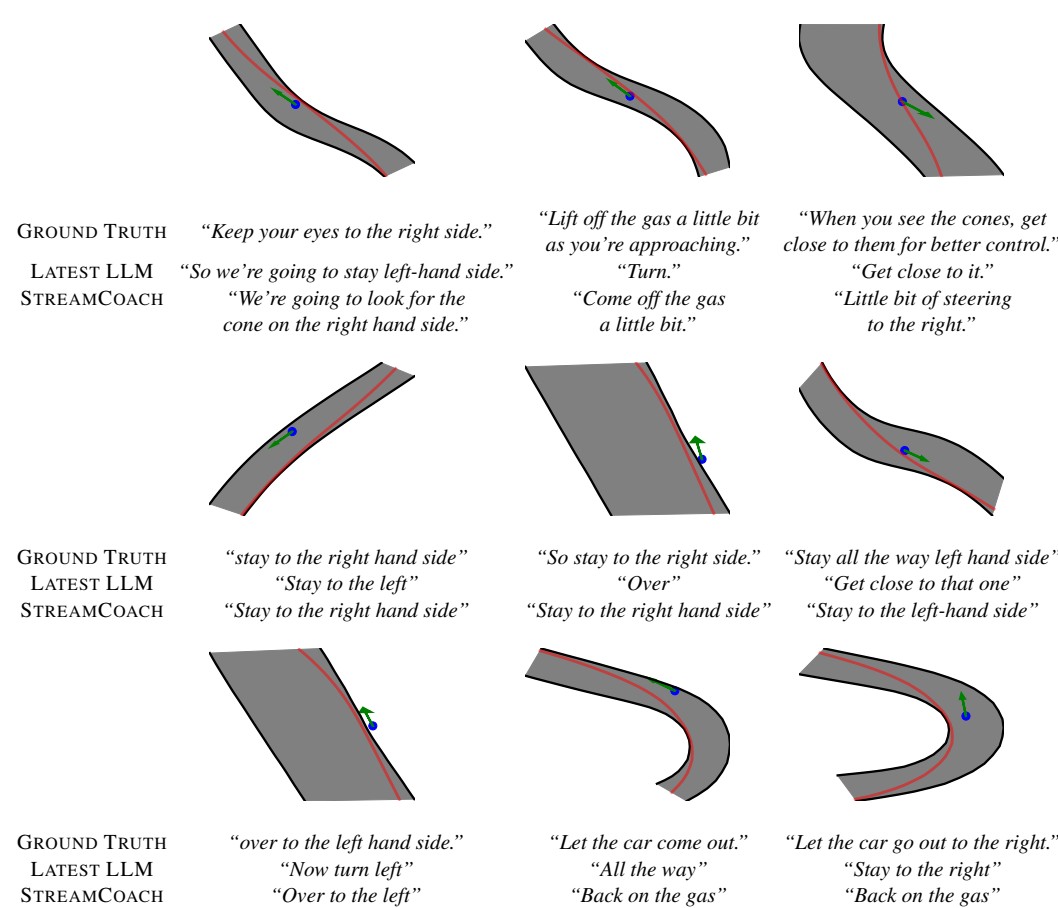

Figure 7: **More Qualitative Results.** The blue dot represents the car's position, the green arrow shows its direction of movement, the red line marks the reference driving line, and the gray line outlines the track border.

Table 8: **Comparison with Long-CoT Reasoning Models for Instruction Generation.**

| MODEL | COSINE SIMILARITY | BLEU | BERTSCORE | METEOR | ROUGE | INFERENCE TIME (SECOND) |
|---|---|---|---|---|---|---|
| GPT-4O-MINI | 0.4767 | 0.1503 | 0.8803 | 0.2994 | 0.3525 | 0.3335 |
| O4-MINI | 0.4152 | 0.0804 | 0.8672 | 0.1966 | 0.2443 | 9.0110 |

The embedding model and reasoning model are critical components of STREAMCOACH. To evaluate their impact, we conducted ablation studies with different configurations for each. For the embedding model, we tested MPNet, *TEXT-EMBEDDING-3-SMALL*, and *TEXT-EMBEDDING-3-LARGE* from OpenAI. As shown in Table 9, consistent with previous findings in (Sinha et al., 2024), the performance across these models was comparable, indicating that larger, commercialized embedding models do not provide significant advantages. In contrast, the reasoning model had a more pronounced impact on performance, as shown in Table 5. We compared three commercialized fast LLMs: GPT-4o-Mini, Gemini 2.0 Flash, and Claude Haiku. GPT-4o-Mini outperformed the other two.

## D  ADDITIONAL LLM AS JUDGE RESULTS

Here, we provide more results using LLM as Judge as the metrics in figure 8. We compare the finetuned version of STREAMCOACH with different number of retrieved examples.

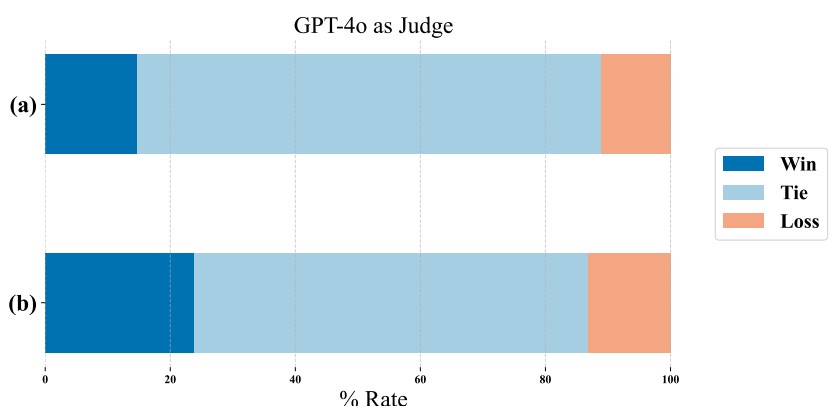

Figure 8: **Additional LLM as Judge Results:** (a) STREAMCOACH (FT, $k$=10) vs. STREAMCOACH (FT, $k$=1), STREAMCOACH (FT, $k$=10) vs. STREAMCOACH (FT, $k$=5)

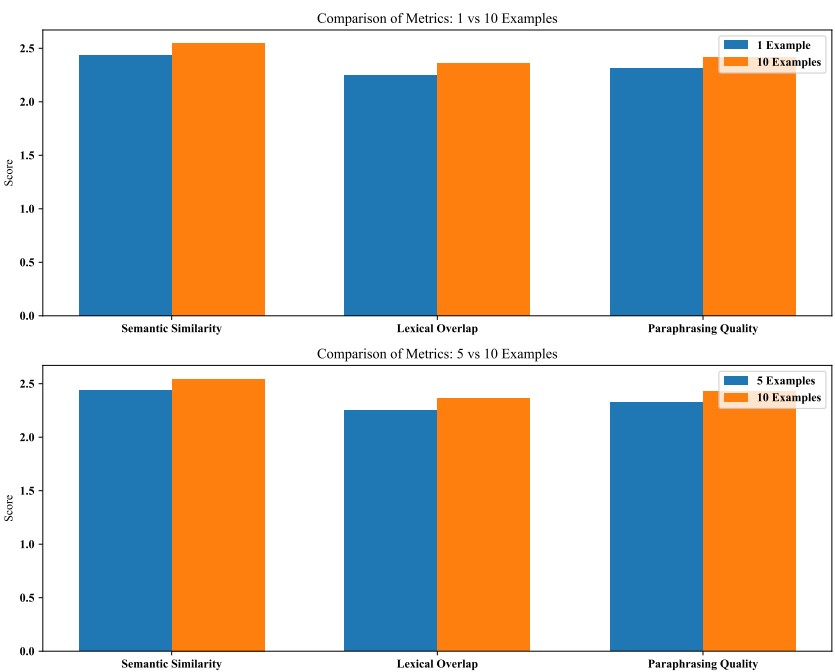

Figure 9: **LLM as Scorer with Rubric Results: Upper:** STREAMCOACH (FT, $k$=10) vs. STREAM-COACH (FT, $k$=1), **Lower:** STREAMCOACH (FT, $k$=10) vs. STREAMCOACH (FT, $k$=5)

In addition to the LLM-as-a-Judge results, we further evaluate the generated instructions using a rubric-based comparison. Specifically, given a pair consisting of a generated instruction and its corresponding ground-truth instruction, we prompt the LLM to assess both according to a set of predefined evaluation metrics:

1. Semantic Similarity – How closely does the candidate convey the meaning of the reference sentence? (score each from 1 to 5)

2. Lexical Overlap – How much lexical content (e.g., key terms or phrases) is shared with the reference? (score each from 1 to 5)

3. Paraphrasing Quality – Does the candidate preserve meaning while using different wording effectively? (score each from 1 to 5)

The result is presented in figure 9 and we put the prompt used in section J. The comparison reveals a consistent trend across all metrics—performance improves slightly as the number of examples

increases. Overall, increasing the number of examples contributes to better paraphrasing performance, but the benefit tapers off beyond a certain point.

## E    INFERENCE TIME VS. PERFORMANCE ANALYSIS

We present a detailed analysis of the trade-off between inference time and performance as a function of the number of retrieved samples ($k$). This analysis is conducted for both prompting-based and fine-tuned models.

**Prompting-based Model.**    We first evaluate the prompting-based version of STREAMCOACH, varying the number of retrieved samples while keeping all other factors constant. All experiments are conducted under identical network conditions and time constraints to minimize the impact of external variables such as bandwidth or server response time. The result is shown in Table 10.

Table 10: **Latency vs. number of retrieved samples for the prompting-based model.**

| #$k$ | COSINE SIMILARITY | BLEU | BERTSCORE | METEOR | ROUGE | INFERENCE TIME (SECOND) |
|------|-------------------|------|-----------|--------|-------|-------------------------|
| 5 | 0.4409 | 0.0719 | 0.8745 | 0.2457 | 0.3048 | 0.5981 |
| 10 | 0.4492 | 0.0852 | 0.8757 | 0.2632 | 0.3223 | 0.5220 |
| 20 | 0.4488 | 0.0938 | 0.8762 | 0.2759 | 0.3326 | 0.5040 |
| 30 | 0.4512 | 0.0927 | 0.8766 | 0.2769 | 0.3352 | 0.5377 |
| 40 | 0.4505 | 0.0977 | 0.8770 | 0.2809 | 0.3410 | 0.5729 |

## F    TRAINING AND INFERENCE DETAILS

In this section, we provide more information about training details of timing and reasoning models.

### F.1    INFERENCE DETAILS

For the prompting-based approach, which relies on a commercial language model, we set the temperature to 0 to ensure deterministic outputs. Since we have limited control over the model's internal behavior and parameters, enforcing determinism helps isolate the effects of our retrieval and prompt design. For the fine-tuned model, we use a temperature of 0.3 to introduce slight variability during decoding. This controlled randomness can improve generalization and output diversity, especially in models we can directly optimize and evaluate across multiple runs.

### F.2    FINETUNED LLM

To fine-tune the LLM using retrieved instructions as input, we employ LoRA with a rank of $r = 32$. Specifically, all fine-tuned models are trained for 4 epochs with an initial learning rate of $2 \times 10^{-4}$, a cosine learning rate schedule, and a warmup phase spanning the first 0.05 epochs using AdamW optimizer.

For the Latest Observation LLM, the input prompts the model to generate the instruction that the assistant is expected to provide based on the latest observation (consisting of three consecutive frames):

> Latest Observation LLM Input
>
> <|begin_of_text|>A multimodal AI assistant is helping coach driver to do car racing in a lap. Below is the stream of state of the ego car, interleaved with the instruction from the assistant.
>
> [;;]
> Assistant:

For Full History LLM and VideoLLM Online, the input would look like:

---

**Full History LLM and VideoLLM Online Input**

<|begin_of_text|>A multimodal AI assistant is helping coach driver to do car racing in a lap. Below is the stream of state of the ego car, interleaved with the instruction from the assistant.

[;;;;;;;
;;;;;;;
;;;;;;;
;;;;;;]
Assistant: A little more gas.

[;;;;;;;
;;;]
Assistant: turn now

[;;;;;;;
;;;;;;;
;;;;;;;
;;;;;;]
Assistant:

The difference in VideoLLM Online lies in whether the LLM is prompted to generate a *;* token, which determines the timing of instruction generation.

For STREAMCOACH, the input would look like:

---

**STREAMCOACH Input**

<|begin_of_text|>A multimodal AI assistant is helping coach driver to do car racing in a lap. Below is the stream of state of the ego car, interleaved with the instruction from the assistant.

[;;;;;;;
;;]
In the similar scenario, instructions given to the driver are: ["A little more gas.", "over to the left.", "turn now,", "over to the left.", "over to the left", "over to the left hand side.", "over to the left.", "get close to that cone.", "over to the left.", "over to the left,"]
Assistant:

---

Here, $\langle s \rangle$ is a special token encoded using a two-layer MLP to represent contextual information. Each contextual input is composed of three $\langle s \rangle$ tokens:

- The first $\langle s \rangle$ encodes *Position* ($\langle x, y, z \rangle$), *Velocity* ($\langle v_x, v_y \rangle$), *Orientation* ($\langle o_x, o_y, o_z, o_w \rangle$) as quaternions, and *Driver's Actions* ($\langle$Steering, Speedometer, Throttle, Brake$\rangle$).
- The second $\langle s \rangle$ encodes map information, including the 20 nearest borders of the track.
- The third $\langle s \rangle$ encodes reference line information, specifically the 20 nearest sample points of the reference line.

For each type of contextual information, we train a separate MLP to encode it into the embedding space, following an approach similar to LLaVa (Liu et al., 2023).

## F.3 CLASSIFIER TRAINING

We construct a MLP neural network as the classifier, with an input size of 768, corresponding to the MPNet embedding size. The network consists of three sequential blocks:

- **First block:** A fully connected layer maps the input (768 dimensions) to 1024 channels, followed by a ReLU activation, and then another fully connected layer maps 1024 channels to 512, also followed by ReLU activation.
- **Second block:** Takes the 512-channel output from the first block and applies two fully connected layers, each maintaining 512 channels. A ReLU activation follows the first layer.

- **Third block:** Maps the 512-channel input from the second block to 256 channels, followed by ReLU activation. It further reduces the size sequentially through 128, 64, and finally 1 channel, with ReLU activations between layers.

The network incorporates a skip connection, where the output of the first block is added to the input of the third block before proceeding through the final layers. This design allows the model to learn residual mappings, improving its ability to capture complex relationships in the data.

The model is trained for 100 epochs using a learning rate of $1 \times 10^{-4}$ and Binary Cross Entropy Loss. A StepLR scheduler is applied, reducing the learning rate by a factor of 0.1 every 30 epochs. To address class imbalance, we adopt a resampling strategy to ensure an equal number of negative and positive samples during training.

## G    SIMULATION ENVIRONMENT AND DATA COLLECTION

In this section, we provide additional details about the simulation environment used for data collection. The simulator runs CARLA (Dosovitskiy et al., 2017) and leverages Robot Operating System (ROS) for hardware integration and for logging vehicle state, controls, video and audio signals. The simulation environment uses Thunderhill West (Willows, CA) track map as the driving circuit. The setup of data collection is shown in figure 10, where the coach is giving instructions to a student who is practicing in the simulation environment.

The overall study session lasted approximately 2 hours and consisted of three main phases:

1. **Familiarization Phase:** The coach introduced the experimental setup, the driving task, and the map layout while performing a sight lap. This was followed by two baseline laps driven by the participant.

2. **Coaching Phase:** As shown in the figure, the coach provided concurrent feedback while the participant drove around the track. After each lap, the coach was given the opportunity to provide additional feedback (terminal feedback). This phase was divided into 15-minute segments. After each segment, participants and the coach completed additional surveys and were checked for signs of motion sickness.

3. **Retention Phase:** Participants completed two laps without any coaching to assess retention of the learned behaviors.

All audio data was transcribed using *Whisper* and subsequently manually corrected and time synced for accuracy. Concurrent feedback was categorized into instruction types using *GPT* via in-context learning, with expert-annotated examples provided as prompts.

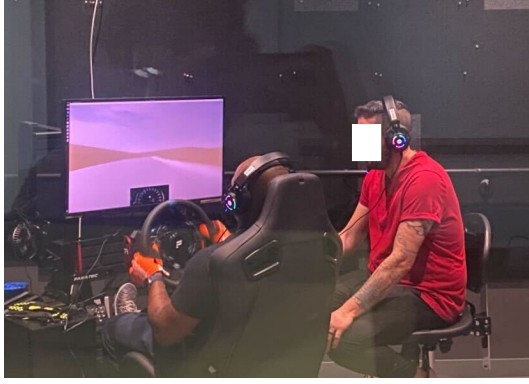

Figure 10: **Data Collection Setup.**

In this setup, the expert delivers frequent (on average every 3 seconds), fine-grained utterances because high-performance driving demands sub-second corrections. Typical guidance includes short, actionable phrases such as "Lift off the gas a little bit," "Stay to the right hand side," or "Let the car

go out to the right," which help the learner adjust line, throttle, and steering continuously. Additional qualitative examples are provided in the appendix (see Appendix A).

This instruction density does not indicate an easy task; rather, it reflects the domain's speed and complexity. The coach must convey what to say and, critically, when to say it as the learner's state evolves non-uniformly over time—motivating our decomposition of fast timing and slow content reasoning. Moreover, because the state space is continuous and language naturally varies (e.g., "Brake now" vs. "Hit the brakes"), we trained with a single, highly calibrated expert to control variability while focusing on learning a coherent coaching policy.

Generating appropriate instructions is challenging even for powerful LLMs with domain context in a low-data regime. As shown in Table 1, baselines struggle even when considering content alone: the *Zero-shot LLM* performs poorly, and models given in-domain examples (*Few-shot LLM*) or fine-tuned on the data (*Latest Observation LLM*; *Full History LLM*) still fail to generate high-quality guidance. The task is further complicated by real-time timing constraints. The system must decide not only what to say but also when to say it, under non-uniformly distributed utterances. The difficulty of solving both problems jointly is highlighted by the *VideoLLM-Online* baseline, which attempts to manage timing and content simultaneously and performs the worst on content while failing almost completely on timing.

Regarding supervision consistency, our dataset was collected with a highly experienced professional coach whose feedback is validated in real-world high-performance driving training. In practice, different coaches—or even the same coach at different times—may provide different instructions for similar situations due to two factors. First, human instruction is inherently stochastic: even if the exact same state could be reproduced, a coach is not a deterministic function. Natural linguistic variation means the same corrective intent might be expressed as "Brake now" or "Hit the brakes," differing in phrasing but not in semantic intent. Second, identical states are rare in this continuous, high-dimensional domain; small differences in speed, trajectory, or throttle produce distinct states that can elicit different corrective feedback (e.g., "turn now" vs. "hold the line") depending on momentary vehicle dynamics.

These factors make learning especially challenging: the model must generalize from a sparse set of unique state–instruction pairs, where each instruction reflects one valid choice among many. As shown in Table 1, baselines such as the *Few-shot LLM* and the *Full History LLM* perform poorly in this setting, lacking mechanisms to handle sparse supervision and expressive variation. By training on a single, highly calibrated expert, we isolate the core challenge of learning a coherent coaching policy before introducing the additional complexity of multi-expert disagreement.

## H State Description

We use description of current frame and one previous frame as the input to the embedding model for retrieval. Here is an example:

---

**Frame Description Example**

Step 1:
The position of the car is: [-663.29, -75.16, 0.94] in meters.
The orientation of the car is: [-0.0, -0.02, 0.7, 0.71] in quaternion.
The velocity of the car is: [0.06, 10.44] in mph.
The speedometer reading is: 23.0 in mph.
The driver's actions are: throttle=0.86 brake=0.0 steering=-0.02
The inner edge of the road is: [[-669.23 -75.52], [-669.47 -73.53], [-669.7 -71.55], [-669.93 -69.56], [-670.17 -67.58]] in meters.
The outer edge of the road is: [[-662.11 -74.86], [-662.35 -72.87], [-662.6 -70.89], [-662.85 -68.9 ], [-663.11 -66.92]] in meters.

Step 2:
The position of the car is: [-663.28, -74.28, 0.96] in meters.
The orientation of the car is: [-0.0, -0.02, 0.7, 0.71] in quaternion.
The velocity of the car is: [0.02, 10.75] in mph.
The speedometer reading is: 24.0 in mph.
The driver's actions are: throttle=0.86 brake=0.0 steering=-0.01
The inner edge of the road is: [[-669.35 -74.52], [-669.59 -72.54], [-669.82 -70.56], [-670.05 -68.57], [-670.29 -66.59]] in meters.
The outer edge of the road is: [[-662.23 -73.86], [-662.47 -71.88], [-662.73 -69.89], [-662.98 -67.91], [-663.24 -65.92]] in meters.

---

# I    PROMPT

Here, we present the prompt used for prompting-based reasoning models. The key difference between models lies in whether the phrase *"In a similar situation, the instructions given were:"* is included in the prompt.

---

### Prompt for Prompting-based Reasoning Models

You are a Driving Coach. You are responsible for providing driving instructions to the driver to learn car racing, here are some instructions you given in some similar situations as reference:

In the similar situation, the instruction have been given are: ["full throttle", "Over to the left,", "A little more gas.", "Over to the left.", "Steer now.", "Over to the left.", "over to the left hand side.", "Over to the left now", "over to the left,", "so small turn to the left", "now turn,", "Now start going over to the left.", "over to the left hand side.", "Now get close to this cone here.", "over to the left.", "a little bit of steering", "over to the left,", "from the right", "Stay to the right,", "over to the left.", ]

Now, The current position of the car is: [-664.59, -47.47, 1.43] in meters.
The current orientation of the car is: [-0.01, -0.02, 0.74, 0.67] in quaternion.
The current velocity of the car is: [-1.87, 17.23] in mph.
The current speedometer reading is: 39.0 in mph.
The driver's actions are: throttle=0.9 brake=0.0 steering=-0.0
The inner edge of the road is: [[-866.23 -455.3 ], [-866.57 -457.24], [-866.82 -459.19], [-867.02 -461.15], [-867.18 -463.11]] in meters.
The outer edge of the road is: [[-874.32 -454.28], [-874.68 -456.36], [-874.98 -458.46], [-875.24 -460.57], [-875.44 -462.69]] in meters.

Inner edge is on the left-hand side and outer side is on the right-hand side. Please provide the next instruction to the driver in a concise way. No more than 10 words. One instruction at once, do not combine. Put your final instruction starting with 'The final instruction is:' without any formatting. Think Step by Step.

---

# J    LLM AS JUDGE PROMPT

Here we present the prompt used for content evaluation under the LLM as judge and LLM as scorer with rubric paradigm. In practice, we ran each comparison twice by switching the order of generated instructions and record the average.

## J.1    LLM AS JUDGE

---

### LLM-as-Judge Prompt

These are two sentences, pick the one that is semantically closer to the reference sentence, output 1 if the first setence is sematically closer, output 2 if the second one is semantically closer. If they are too similar with each other or both different from the reference semantically, output 0, the reference is: Take the left turn

1. Over to the left
2. Head over to the cone

---

## J.2 LLM AS SCORER WITH RUBRIC

---

**LLM-as-Scorer Prompt**

You are given a reference sentence and two candidate sentences. Your task is to evaluate each candidate sentence independently using the rubric below, and provide a score for each criterion. Do not choose a winner, simply assess both candidates.

Rubric (score each from 1 to 5):
1. Semantic Similarity: How closely does the candidate convey the meaning of the reference sentence?
2. Lexical Overlap: How much lexical content (e.g., key terms or phrases) is shared with the reference?
3. Paraphrasing Quality: Does the candidate preserve meaning while using different wording effectively?

Reference sentence:
"Take the left turn"

Candidate 1:
"Over to the left"

Candidate 2:
"Head over to the cone"

Output Format (JSON):
{
"candidate_1": {
"semantic_similarity": X,
"lexical_overlap": X,
"paraphrasing_quality": X
},
"candidate_2": {
"semantic_similarity": X,
"lexical_overlap": X,
"paraphrasing_quality": X
}
}
(Replace X with scores from 1 to 5, where 5 is best.)

---

## K  DETAILED PROMPTING PROCEDURE

We first encode the **language-based descriptions of environment states** from the training set where expert instructions were provided, using a pretrained sentence embedding model $\phi(\cdot)$. Specifically, each observation $o_t^*$ with a non-empty expert instruction $\mathcal{I}_t^* \neq \emptyset$ is translated into a language description and mapped to an embedding $e_t^* = \phi(o_t^*)$, which is stored in the positive retrieval cache $D_{\text{positive}}$.

At inference time, the current observation $o_t$ is similarly converted into a language description using the same predefined template and embedded as $e_t = \phi(o_t)$. We then compute the cosine similarity between $e_t$ and each embedding $e_t^* \in D_{\text{positive}}$ as:

$$\text{sim}(e_t, e_t^*) = \frac{e_t^\top e_t^*}{\|e_t\| \cdot \|e_t^*\|}.$$

The **top-**k most similar embeddings, denoted as $\{e_{t_1}^*, e_{t_2}^*, \ldots, e_{t_k}^*\}$, are selected based on cosine similarity. For each retrieved embedding $e_{t_i}^*$, we retrieve the corresponding expert instruction $\mathcal{I}_{t_i}^*$. These $k$ retrieved instruction-context pairs are then aggregated to form a composite prompt $P_t$ that includes both the retrieved examples and the current state description.

This prompt $P_t$ is passed to the LLM within a **retrieval-augmented generation (RAG)** framework to produce a new, context-sensitive instruction:

$$\mathcal{I}_t = R(P_t),$$

where $R(\cdot)$ denotes the reasoning model (either prompting-based or fine-tuned) that generates free-form language instructions.

## L   THE USE OF LARGE LANGUAGE MODELS

Beyond their use described in the main text, LLMs were also employed to refine the writing of this paper

