# OpenReview forum: "Real-Time Coaching of Human Physical Skills with Large Language Models"
_ICLR.cc/2026/Conference — ICLR 2026 Conference Withdrawn Submission_

### Official Review · Reviewer_zXdc · 2025-10-16

**Soundness:** 2
**Presentation:** 3
**Contribution:** 3
**Rating:** 6
**Confidence:** 3

**Summary:**

This paper presents StreamCoach, a two-stage coaching framework for concurrent coaching: a fast inference stage for deciding when to intervene and a slow reasoning stage for deciding what to say. The two stages are bridged by a shared representation space. The problem is practical for high-stakes, real-time domains. StreamCoach outperforms existing baselines in both intervention timing and instruction quality.

**Strengths:**

1. The definition and decomposition of the problem is clear that splitting *when to intervene* from *what to say* and using a shared representation space for practical implementations.
2. StreamCoach delivers consistent improvements in both intervention timing and instruction quality against baselines.

**Weaknesses:**

1. Despite multiple metrics for measuring content similarity, the quality of generated instructions is mainly judged by GPT-4o, which may introduce bias. The experiments section can significantly benefit from a user study to verify the effectness of methods.
2. The experiments provide the sufficient validation for the claims of the paper. However, I believe that the limitations of the method are not properly addresses.
3. In some places you use "LLM as Judge" (line 914, 932) and elsewhere "LLM-as-judge" (line 959). Please corret and standardize the phrasing.

**Questions:**

1. Do the results of human evaluation align with the metrics and LLM-as-judege evaluations?
2. Is visual input alone sufficient for SteamCoach to make immediate intervention decisions and for students to take action in real-world deployments, or should the coach incorporate prior knowledge (e.g., track maps, car speed, weather) to anticipate upcoming situations?
3. What is the common failure cases of StreamCoach? How does the system behave under a false alarms or incorrect instructions?

---

### Official Review · Reviewer_7c39 · 2025-10-19

**Soundness:** 1
**Presentation:** 2
**Contribution:** 2
**Rating:** 2
**Confidence:** 4

**Summary:**

This paper introduces STREAMCOACH, a two-stage framework for real-time coaching of human physical skills. Both stages build on embedding similarity matching. A fast inference stage decides when to intervene by comparing the current (language-templated) state embedding against caches of positive/negative examples and a binary classifier, while a slow reasoning stage decides what to say by retrieving top-k similar positive examples and using RAG to generate free-form instructions. On a driving task in CARLA (Thunderhill West track), the experiment demonstrates real-time timing decisions (with embedding extraction taking 17 ms) and improved instruction quality and timing compared with existing baselines. Quantitative results (content metrics, timing metrics, and an overall "teaching score" = timing × content) favor STREAMCOACH; ablations probe retrieval size, cache size, time window, and reasoning model choice.

**Strengths:**

- The practical problem settings and experiments convincingly demonstrate the framework’s real-time capability (17 ms for intervention timing and 350 ms for instruction generation), showing a tangible step toward achieving reasoning speeds comparable to human reaction time in the context of real-time coaching task.
- The work is easy to follow, with a simple retrieval-based process that enables both validation and test-time replanning.
- The paper clearly articulates the key properties of the real-time coaching problem, as balancing the trade-off between accuracy and latency remains a challenging and underexplored issue.
- The results support the claims made in the paper, and the dataset collection process is transparently reported.

**Weaknesses:**

- The method shows heavy reliance on the embedding-based inference (classifier) performance, where even a slight domain shift could become a bottleneck for the coaching system. Over-triggered re-planning may also cause unnecessary computational overhead.
- The paper lacks an analysis of the scale and diversity of skill sets that require real-time coaching, such as physical or motor skills. Retrieval performance would likely vary depending on the number of data points per skill set, yet this aspect is not discussed.
- The experiments primarily report coaching accuracy and timing metrics, but do not evaluate task-level performance in the actual autonomous driving scenario. This raises concerns about practical effectiveness. For instance, even suboptimal trajectories might still lead to success, while a single incorrect coaching intervention (e.g., during a left turn) could cause catastrophic failure.
- Since the reported performance relies entirely on the composite metric $R_t = r_t^{timing} * r_t^{content}$, the evaluation design raises concerns, especially regarding the use of cosine similarity to define $r_t^{content}$. High cosine similarity does not always reflect semantically accurate or contextually appropriate guidance. [1,2,3]
- For example, the instructions could have similar embeddings yet lead to contradictory actions with longer instruction.
    - *[ground truth] “Stay to the **right hand side**”*
    - *[Instruction A] “We're going to stay in the **right hand side** of the racetrack here”*
    - *[Instruction B] “Stay to the **left hand side**”*

    Using OpenAI’s text-embedding-3-large model, I computed cosine similarities between 3072-dimensional instruction embeddings. Instruction B, despite having the opposite meaning of the ground truth, yielded a higher similarity score.

    |  | *ground truth* | Instruction A | Instruction B |
    | --- | --- | --- | --- |
    | *ground truth* | 1.000 | 0.666 | 0.937 |
    | Instruction A | 0.666 | 1.000 | 0.633 |
    | Instruction B | 0.937 | 0.633 | 1.000 |
- The experiments are conducted using a single simulator, a single track, and a single expert, which limits the assessment of the proposed approach’s generalization capability. Although the paper includes an appendix discussion on evaluating cases where multiple coaching instructions share the same meaning but differ in expression, it provides limited evidence from integrated experiments demonstrating the method’s generalization capability.

[1] Ettinger, Allyson. “What BERT Is Not: Lessons from a New Suite of Psycholinguistic Diagnostics for Language Models.” Transactions of the Association for Computational Linguistics, vol. 8, 2020

[2] Zhou, Kaitlyn, et al. “Problems with Cosine as a Measure of Embedding Similarity for High Frequency Words.” Proceedings of the 60th Annual Meeting of the Association for Computational Linguistics: Short Papers, 2022

[3] Weller, Orion, et al. "On the theoretical limitations of embedding-based retrieval." arXiv preprint arXiv:2508.21038, 2025

**Questions:**

- Could the authors provide clearer motivations for why the coaching task must operate in real time, and explain under what conditions real-time responsiveness meaningfully affects learning outcomes or user performance?
- What happens to STREAMCOACH when the ground-truth instructions become longer or noisier?

---

### Official Review · Reviewer_SzD6 · 2025-10-28

**Soundness:** 2
**Presentation:** 3
**Contribution:** 2
**Rating:** 4
**Confidence:** 4

**Summary:**

This paper studies concurrent coaching of physical skills and introduces STREAMCOACH, a two-stage “fast–slow” framework that separates when to intervene from what to say  using large language models. The primary application and evaluation domain is high-performance race car driving. The authors evaluate STREAMCOACH in the CARLA simulator, demonstrating that it significantly outperforms existing baselines in both intervention timing and instruction quality.

**Strengths:**

* The paper tackles a significant and challenging problem: providing real-time, language-based coaching for high-speed physical skills. The proposed fast-slow decomposition is an elegant and practical solution to the inherent tension between latency and generation quality that plagues many real-world LLM applications.

* The authors benchmark STREAMCOACH against a broad spectrum of strong baselines (rule-based heuristics, few-shot prompts, fine-tuned language models, and end-to-end reinforcement-learning agents), thereby providing a clear picture of where their approach excels and where it remains limited.

*  The paper is well-written and easy to follow. The problem statement is clearly articulated, and the proposed solution is explained with clarity.

**Weaknesses:**

* External validity limitations. The experiments are conducted within a single domain (driving), on a single track, and using data from a single expert coach, results may overfit a narrow domain. While the authors justify using a single expert to ensure data consistency, this raises questions about the model's ability to generalize. How would the framework handle different tracks, different skills (e.g., drifting vs. racing), or different coaching styles? More importantly, how would it resolve conflicting advice if trained on data from multiple experts? A discussion of the potential challenges and strategies for generalization would strengthen the paper.

* Decision-rule inconsistency and unused thresholding. The paper presents conflicting trigger rules (e.g., ''generate if $s_{\text{pos}} < s_{\text{neg}}$ or $f = 1$'' in section 4.1  vs $\Delta s = s_{\text{pos}} - s_{\text{neg}}$ in Alogorimthm 1). A threshold $\tau$ is listed in Algorithm 1 but unused in equations.

* Content is judged mainly by textual similarity and an LLM judge. There is no explicit metric for action consistency (e.g., brake/accelerate/steer direction/strength), or behavioral gains tied to instructions (off-track rate, smoothness). This weakens claims about effective coaching.

* Absence of Human-in-the-Loop Evaluation. The ultimate goal is to accelerate human skill acquisition. A small-scale user study, even with a limited number of participants, comparing learning improvement with STREAMCOACH versus a baseline or no coaching , would be valuable.

**Questions:**

* Generalization. How does STREAMCOACH transfer across tasks or to a multi-expert setting? Could you elaborate on how you envision adapting STREAMCOACH so that the RAG framework can retrieve and synthesize different coaching styles?

* To solidify the claim that the system delivers effective coaching, please include action-level executability metrics: quantify consistency (did the agent execute the intended maneuver type and parameters?) and behavioral gains (lap-time reduction, off-track or rule-violation rate, steering/brake smoothness).

* Could the authors elaborate on their plans for or thoughts on a human-in-the-loop user study? I believe it would provide much more direct and compelling evidence of the system’s practical utility.

---

### Official Review · Reviewer_K8tN · 2025-10-30

**Soundness:** 2
**Presentation:** 3
**Contribution:** 2
**Rating:** 2
**Confidence:** 3

**Summary:**

This paper introduces StreamCoach, a novel framework for automating real-time coaching in human physical skill learning, particularly high-speed driving. While prior approaches have attempted to jointly address both when to intervene and what to say, they often face a trade-off between real-time responsiveness and content quality. In contrast, this work decouples the two tasks into distinct stages. In the fast inference stage, the learner’s state is encoded into language embeddings, enabling intervention decisions within 17ms. In the slow reasoning stage, Retrieval-Augmented Generation (RAG) is employed to produce contextually grounded and personalized instructions. Experiments conducted in the CARLA high-performance driving simulator demonstrate that StreamCoach outperforms existing systems in both intervention timing accuracy and instruction quality.

**Strengths:**

- Clear problem definition and decomposition: The key contribution lies in decomposing real-time coaching into two subtasks, timing decision and language generation. This modular design effectively resolves latency issues observed in end-to-end models, while closely emulating how human coaches decide when and what to say.

- Technical reproducibility: The paper provides detailed accounts of data collection, model architecture (embedding, classifier, RAG pipeline), and training hyperparameters. In addition, the appendix includes prompt templates and LLM evaluation rubrics, greatly enhancing reproducibility.

- Comprehensive experimental evaluation: The study evaluates StreamCoach across multiple quantitative metrics, including timing accuracy, content similarity, and an overall teaching score, and conducts ablation studies to analyze the contribution of each design component.

**Weaknesses:**

- Evaluation Scope: The evaluation focuses primarily on instructional alignment, measuring timing accuracy and linguistic similarity with expert demonstrations. However, it does not examine whether such coaching translates into improved learner behavior or task performance, which is crucial for establishing effective concurrent coaching.

- Objective Definition: The learning objective is formulated as reproducing expert-issued interventions rather than optimizing end-task performance. For broader applicability in sequential decision-making domains such as autonomous driving or sports, the framework should clarify how the coaching signal contributes to goal-directed skill improvement.

- Generalization Limitation: Experiments rely on data from a single expert coach within one simulated environment, making it unclear how StreamCoach generalizes to diverse feedback styles or domains. Multi-expert or cross-domain validation would strengthen claims of scalability.

- Input Modality Constraint: The system represents learner states as language descriptions for embedding efficiency and interpretability, but lacks vision-language grounding. As acknowledged in the Conclusion, this constrains real-world applicability where multimodal perception is essential.

- Behavioral Validation: No behavioral task validation is conducted to assess whether AI-generated coaching actually accelerates skill acquisition or improves decision quality. Such validation would substantiate the claim of effective and scalable real-time coaching.

**Questions:**

- How does alignment with expert timing and phrasing translate into actual performance improvement for learners?

- Would integrating a performance-based reward alongside expert imitation help StreamCoach better optimize for learning outcomes rather than similarity?

- How sensitive is the framework to variation in coaching style or task domain? Could the embedding space or RAG retrieval cache be adapted for multi-expert or cross-domain coaching?

- Since the paper notes the lack of visual inputs as a limitation, how might future extensions incorporate vision-language embeddings or sensor fusion to enhance contextual grounding?

- Has any behavioral task validation been conducted to determine whether AI-generated coaching actually accelerates skill acquisition or improves decision quality?

---

### Note · Authors · 2025-11-21

I have read and agree with the venue's withdrawal policy on behalf of myself and my co-authors.